# Factors of Compliance of Dental Patients in Primary Health Care Services during the Pandemic

**DOI:** 10.3390/vaccines11040844

**Published:** 2023-04-14

**Authors:** Sofia Zachari, Dimitrios Papagiannis, Ourania Kotsiou, Foteini Malli, Evangelos C. Fradelos, Konstantinos I. Gourgoulianis

**Affiliations:** 1Primary Health Care Post Graduate Program, University of Thessaly, 41110 Larissa, Greece; esofia.zahari@hotmail.gr; 2Public Health & Vaccines Laboratory, Department of Nursing, School of Health Science, University of Thessaly, 41500 Larissa, Greece; 3Human Pathophysiology Laboratory, Department of Nursing, University of Thessaly, 41500 Larissa, Greece; raniakotsiou@gmail.com; 4Respiratory Disorders Laboratory, Faculty of Nursing, University of Thessaly, 41500 Larissa, Greece; mallifoteini@yahoo.gr; 5Department of Nursing, University of Thessaly, 41500 Larissa, Greece; evagelosfradelos@hotmail.com; 6Respiratory Medicine Department, Faculty of Medicine, University of Thessaly, 41110 Larissa, Greece; kgourg@med.uth.gr

**Keywords:** dental patients, vaccination, COVID-19, compliance, primary health care

## Abstract

The compliance of dentists with good hygiene practices during the COVID-19 pandemic was essential to minimize the transmission of SARS-CoV-2 strains, and the pandemic has significantly interrupted the provision of oral health care to many individuals. We aimed to examine, in a cross-sectional study, factors affecting dental patients’ compliance in primary dental health settings during the pandemic period. The present study was conducted on 300 dental patients visiting four private dental offices in the city of Larissa in Central Greece during the period October–December 2021. The patients in the study sample were of an average age of 45.79 years with a standard deviation of 15.54 years, and 58% of the sample were females. A significant proportion of the participants, 22%, reported that they would be influenced if they knew that the dentist had been ill with COVID-19 despite being fully recovered. A total of 88% of the participants reported that they would feel safe if they knew their dentist was vaccinated against COVID-19. Regarding the information received by dentists, 88% of the participants agreed that the dentist’s role is important in dealing with the COVID-19 pandemic, and 89% of them agreed that the information they received from the dentist about the COVID-19 pandemic was sufficient. One-third of the total sample reported that COVID-19 negatively impacted keeping dental appointments, and 43% of the sample kept scheduled appointments. A total of 98% reported that the dentist followed all health protocols against COVID-19 and that their office was equipped to follow health protocols. In the present study, we observe that dentists had adequate knowledge of, attitudes towards, and practices of infection control protocols against COVID-19 during the second wave, according to patients’ perceptions.

## 1. Introduction

The outbreak of the novel coronavirus SARS-CoV-2 led to global disorganization, with the World Health Organization announcing its spread as a pandemic in early March 2020 [1]. During the pandemic, dentists faced enormous challenges regarding the restrictions placed on their practice and had to comply with biosafety measures. Many cases of infection in healthcare professionals are attributed to aerosol-generating procedures performed in high-risk areas, such as intensive care units, emergency rooms, respiratory medicine settings, and dental clinics. In a dental setting, the risk of transmission is significantly increased as aerosol and droplet production from dental instruments, coupled with the patient’s secretions, saliva, and blood, potentially lead to an increased risk of pathogenic microorganism spread [2]. The face-to-face contact between patients and dental care workers (DCWs) in the dental clinic represents a high-risk condition of transmission [3].

The mechanism of transmission by aerosol-generating procedures (AGPs) is different from that of droplet transmission as it involves the presence of microbes within droplet nuclei, which are generally considered to be particles of <5 μm in diameter, that can remain in the air for long periods of time and be transmitted to others over distances greater than 1 m. Aerosol-generating procedures (AGPs) in dental practice remain a health concern, since aerosols produced during clinical procedures can be contaminated with microorganisms, which can cause respiratory health effects or transmit diseases bidirectionally among dentists, dental health care workers and patients [4].

Threat appraisal is considered one element that influences health behaviors. The primary health care model suggests that health behaviors, such as attending dental appointments, are controlled by individual perceptions, and modifying factors [5]. Many studies were conducted on the relationship between preventive behavior and risk perception in the general population during the COVID-19 pandemic [6,7]. Maintaining high hygiene standards in a dental clinic and adherence to infection prevention protocols are essential to achieving the desired goal. Dental staff must be at the forefront of cross-infection control to protect patients’ well-being and oral health. Informed clinical decisions must be made; public awareness is raised to avoid panic while promoting patients’ oral health and well-being in these challenging times. In 2020, a global study was carried out to assess dentists’ knowledge of, attitude towards, and practices with the new data of the pandemic. According to these results, all dentists (100%) agreed that it was possible through their work to participate in raising the awareness of patients about COVID-19 and that hand hygiene and personal protective equipment were very effective in preventing contamination [8].

The aim of the present study was to examine the relationship between the perceived susceptibility of patients with dental disorders regarding the implementation of protective measures and if the vaccination status or other factors of the dentists during the COVID-19 pandemic affected the compliance of dental patients in primary health care services during the pandemic.

## 2. Materials and Methods

A brief cross-sectional survey was conducted on dental patients visiting four private dental offices in the city of Larissa in Central Greece. A self-report questionnaire with questions on demographics and 8 questions on attitude/perceptions was used as a research tool, having been drawn up after a review of the relevant literature. The questionnaire included closed-ended questions divided into the following sections: (a) demographics that investigated the socio-demographic information of the participants, including gender and residence and (b) 8 questions on attitudes towards and perceptions of the implementation of preventive measures against COVID-19 and the role of dentists during the pandemic. The questionnaire was initially distributed to 10 patients whose comments were used to shape the final version. In contrast, its final version was finalized after being examined by an experienced university researcher (ensuring content and face validity, respectively). The research was carried out during the period October–December 2021. in accordance with the Declaration of Helsinki, verbal informed consent was obtained from all participants prior to entering the study, as well as approval from the University of Thessaly’s ethics committee. The inclusion criteria were an age of over 18 and the provision of consent to be interviewed for the study. The main research questions were about the impact of the pandemic on the dental care of the patients, how the prevention measures, such as the vaccination status of dentists, affect their behavior, and what the nature of their compliance was with the dental services in primary health care during the pandemic period.

### Statistical Analysis

Descriptive and inferential statistical analyses were performed. Relative (%) and absolute frequencies were presented for categorical variables, while continuous variables were presented as mean ± standard deviation. The chi-squared test (χ^2^) was used for the univariate analysis of categorical variables; the *t*-test was used for the univariate analysis of continuous variables. A *p*-value of <0.05 was considered indicative of statistical significance. Logistic regression with the backward method was performed to identify which variables impact the decision to attend an appointment with a dentist. All data were analyzed with the SPSS software (IBM SPSS Statistics for Windows, Version 26.0. Armonk, NY, USA: IBM Corporation).

## 3. Results

Among the 450 patients, 300 were accepted to participate. A total of 30 participants had missing values, and 120 participants refused to participate. The age of the subjects ranged from 18 to 86 years, and the mean age of participants in the study sample was 45.79 years with a standard deviation of 15.54 years. Among the participants included, the median (IQR) age was 45 (an inter-quartile range (IQR) of 34–55.25) and the gender distribution was that 58% of the total sample was female.

The majority of the participants were married (62%), two-thirds of the sample had at least one child (68%), and the vast majority of them (85%) had a monthly income of less than 1500 euros. In terms of residence, 78% of the population lived in urban areas, and the rest lived in rural areas (Table 1).

Regarding the education level of the participants, two-thirds 65% were university graduates and postgraduates while 8% were primary school graduates. A total of 24% of the sample canceled planned prosthetic work (dentures, implants, or crown placements) due to the fear of the transmission of COVID-19, while 46% did not undergo teeth cleaning (Table 2).

In the question about the compliance with teeth cleaning in the previous year, 54% of the participants reported that they visited the dental settings for teeth cleaning. Meanwhile, the majority reported that they planned to cancel prosthetic work (dentures, implants, or crown placements) due to the fear of the transmission of COVID-19 (Table 2). The vast majority of the participants recognized the important role of dentists in primary health care against the pandemic. In the question “Has COVID-19 discouraged you to keeping your scheduled dental appointments?” 67% reported that they had not been affected.

The prevalence of COVID-19 among participants was 45%. An significant proportion of the participants, 22%, reported that they would be influenced if they knew that the dentist had been affected by COVID-19 despite being fully recovered. The vast majority of participants, 88%, reported that they would feel safe if they knew their dentist was vaccinated against COVID-19. Regarding the information received by dentists, the vast majority of the participants, 88%, agreed that the dentist’s role is important in dealing with the COVID-19 pandemic, and 89% of them agreed that the information they received from the dentist about the COVID-19 pandemic was sufficient. One-third of the total sample, 33%, reported that COVID-19 negatively impacted keeping dental appointments, and 43% of them kept scheduled appointments. The vast majority of participants, 98%, reported that the dentists followed all health protocols against COVID-19 and that their office was equipped to follow health protocols (Table 2). The patients who were sick kept their appointments at a significantly higher rate than those who were not sick (χ^2^ = 11.206, *p* = 0.010).

A logistical regression with the backward method was performed in order to identify which variables can affect the decision of patients to attend appointments with a dentist. According to the regression analysis, the prognostic factors that play an important role in patients keeping their appointments are the perception that dentist have an important role in the pandemic (OR = 0.654), the information regarding COVID-19 that dentists give to their clients (OR = 1.502) and the illness and recovery status of dentists (OR = 1.773), (Table 3). Of all the dentists and dental health care workers who participated in the present study, 100% were fully vaccinated against COVID-19.

## 4. Discussion

In the present study, it is shown that the COVID-19 pandemic significantly impacted the provision of dental care, with almost a third of patients postponing or worrying about keeping their appointments. Patients and dentists experienced the consequences of the pandemic on their health and professional-financial level. At the health facilities, one of the most effective measures was classifying services as essential or non-essential, following guidelines by organizations or countries’ public health authorities, which allowed resources to be redirected to the pandemic response. Nevertheless, this also caused cancellations or delays in elective and non-urgent procedures [9,10,11].

In the international literature, there are a huge number of scientific surveys that have been published and focus on the impact of the pandemic on different aspects, including, among others, the use of certain specific services, such as maternal and child health care, child vaccination, or chronic diseases in the initial stages of the pandemic [12,13,14]. A systematic review by Alqahtani et al. reported a progressive reduction in hospital admissions for COPD exacerbations associated with the COVID-19 pandemic, with the calculated percentage of reduction varying between 27% and 78% [15]. In the present study, we report that only 24% of the participants who were not sick kept their scheduled appointments at dental offices.

Health policies, programs, or interventions can, in turn, affect access barriers related to health services or change the mutable characteristics of a community. A key measure in the fight against COVID-19 is vaccination. Vaccination programs were launched worldwide in December 2020, with many countries having now vaccinated a substantial proportion of their population. Vaccination is vital in protecting the public against COVID-19. In the present study, one of the reasons for keeping appointment withs dentists was the vaccination status of the dentists. In Greece, the vaccination coverage for health professionals was very high. A previous study in Central Greece among health professionals indicates a high level of COVID-19 vaccination acceptance among physicians, dentists, and pharmacists [16].

The high acceptance of vaccination against COVID-19 among dentists and the consequently favorable acceptance of protection by their clients positively impact the population in terms of increasing awareness and vaccination. In addition, participants who had moderate and high levels of knowledge of vaccination against COVID-19 were more likely to accept receiving the vaccine, and participants whose level of fear of COVID-19 was high were more likely to accept receiving the vaccine compared to those with a low level of fear [17]. In a study conducted in the Czech Republic, 79.6% of dentists surveyed stated they were fully vaccinated. The more positive attitude to vaccination against COVID-19 among dentists could be attributed to their type of training [18]. Abedin et al. showed that a general population with a higher level of education has a more positive attitude towards vaccination [19].

One of the most crucial reasons for enhanced vaccine hesitancy in the community is the rejection of vaccination by health professionals. Ensuring the acceptance of vaccines, especially by health professionals, is just as important; several studies investigated the willingness to take a potential COVID-19 vaccine or vaccination [20,21,22]. In the present study, we report that the vaccination of dentists played a pivotal role in reducing the fear of patients visiting the dental office during the second pandemic wave. In the present study, the vast majority of participants, 88%, reported that they would feel safe if they knew their dentist was vaccinated against COVID-19.

Another critical issue indicated in the pandemic period was how sufficient the information that patients received from professionals was. The present study, in accordance with other studies, reported that health professionals played a vital role in the education and integration of health and social services during the COVID-19 pandemic [23,24]. A distinctive approach to infectious diseases compared with other conditions is fear. Fear is directly associated with the transmission rate of COVID-19 and its moderately quick and invisible spread, as well as its morbidity and mortality. Fear also leads to psychosocial challenges, including stigmatization, discrimination, loss of life, or the cancellation of appointments in primary health care facilities [25,26] According to Schepers et al., appointments were more likely to be canceled by the patient than the provider and more likely to be canceled by medical specialists such as dentists or ophthalmologists rather than general practitioners [26]. In the present study, we report that 76% of the participants did not cancel planned prosthetic work (dentures, implants, or crown placements) due to the fear of the transmission of COVID-19, and 24% canceled their planned appointments due to the fear of COVID-19 transmission. Compared with the present results, a fear of infection reduced the willingness to see a physician, causing missed medical appointments in the first pandemic wave. According to another review from different disciplines and areas, the number of missed medical appointments substantially increased during the first wave of the COVID-19 pandemic in early 2020 [27].

The Greek Public Health Organization (EODY) suggests hand washing as a mandatory step before and after a dental procedure, before leaving the dental office, after touching anything contaminated by blood, saliva, or other secretions, and when one’s hands are visibly soiled. In the present study, the participants agreed that the vast majority of dental professionals, 94%, showed good knowledge of hand hygiene and followed all health protocols against COVID-19 according to the guidelines of public health authorities while carrying out dental work on patients. Our results agree with those of other studies [28] and are the opposite to the findings of other studies [29,30].

The present study has several limitations. It was carried out during the third wave of the pandemic when there had already been some degree of familiarization among patients and dentists with the increased protection measures. The present study is descriptive, and due to its nature, it is not possible for the authors to provide the reader with causal associations between investigated risk factors and the compliance with primary healthcare services during the pandemic. Due to the questionnaire-based structure of our study, recall or information bias may have occurred. The convenience sampling of the participants is another limitation, and the small number of dental office participants needed to be representative of the dental community.

## 5. Conclusions

In the present study, we reported that dentists had adequate knowledge of, attitudes towards, and practices of infection control protocols against COVID-19 during the second wave, according to patients’ perceptions. The vast majority of dental professionals showed good knowledge of hand hygiene and followed all health protocols against COVID-19 according to the guidelines of public health authorities while carrying out dental work on patients. The patients sometimes canceled planned work due to the fear of the transmission of COVID-19, and the vaccination status of dentists played a pivotal role in reducing the fear of patients visiting the dental office in the second year of the pandemic. The vast majority of patients reported that they would feel safe if they knew their dentist was vaccinated against COVID-19. According to the international guidelines for the novel coronavirus, dental practitioners are recommended to adopt protective measures to avoid respiratory infections that can be transmitted through droplets of different sizes [31]. Our results support the implementations of these guidelines and that they should be followed by the dentists. Despite the limitations, these results are valuable in light of many studies on the behavior of dental patients during the pandemic period. In the future, based on the present study results, we will find it necessary to elaborate more specific criteria for assessing the personality characteristics of patients which determine their compliance with dental work in dental health offices and the implementation of hygiene practices after the experience of the pandemic.

## Figures and Tables

**Table 1 vaccines-11-00844-t001:** Demographic characteristics of the participants.

Demographics	n, %, Mean Value	
Gender		
Female	174	58
Male	126	42
Marital status		
Married	188	62
Single	98	32
Divorced	14	6
Children		
Yes	204	68
No	96	32
Number of children		
0 children	1	
1 child	47	
2 children	103	
3 children	40	
4 children	12	
Monthly Income		
<500 €	34	11
500–999 €	112	38
1000–1499 €	109	37
1500–1999 €	25	8
>2000 €	20	6
Residence		
Rural area	62	22
Urban area	238	78
Education		
Primary	26	8
Secondary	83	27
Graduate	133	44
Postgraduate	59	19

**Table 2 vaccines-11-00844-t002:** Dental patients and their compliance with primary health care services during the period of COVID-19 pandemic.

Questions	N %	
Did you undergo teeth cleaning during the last year?		
Yes	163	54
No	137	46
Did you cancel planned prosthetic work (dentures, implants, crown placements) due to the fear of the transmission of COVID-19?		
Yes	71	24
No	229	76
The dentist’s role is important in dealing with the COVID-19 pandemic?		
Strongly agree	110	37
Agree	155	52
Strongly disagree	6	2
Disagree	29	9
The information you received from the dentist about the COVID-19 pandemic sufficient?		
Strongly agree	144	48
Agree	124	41
Strongly disagree	5	2
Disagree	27	9
Has COVID-19 discouraged you to keeping your scheduled dental appointments?		
Yes	100	33
No	200	67
You consider for your appointments whether the dentist follows all health protocols against COVID-19 according to guidelines of public health authorities.?		
Strongly agree	223	74
Agree	72	23
Strongly disagree	1	1
Disagree	4	2
If you knew that the dentist he had fallen ill from COVID-19 and fully recovered, would it affect your visit to the dental office?		
Yes	67	22
No	233	78
Would you feel safe knowing that your dentist was vaccinated against the COVID-19 disease?		
Yes	264	88
No	36	12

**Table 3 vaccines-11-00844-t003:** Multiple logistic regression results with willingness to keep appointments as the dependent variable.

Results of Multiple Logistic Regression with Willingness to Keep the Appointment as Dependent Variable						
	Full Model		Reduced Model			
	Odds Ratio	*p*-Value	Odds Ratio	CL 95%		*p*-Value
Important role of dentists against the pandemic	0.65	0.02	0.65	−0.79	−0.06	0.02
Information about COVID-19 from dentists	1.42	0.06	1.50	0.07	0.75	0.01
Illness of dentist and patient’s visit	1.70	0.07	1.77	0.01	1.14	0.04
Constant	0.32	0.04	0.47	−1.40	−0.12	0.02

## Data Availability

Data are available upon request.

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
