# Peer review of "Factors of Compliance of Dental Patients in Primary Health Care Services during the Pandemic"

_vaccines, 2023, doi:10.3390/vaccines11040844_

Round 1

Reviewer 1 Report

see attached file.

Author Response

Reviewer 1

This is an interesting and timely paper. There are, however, some items that need clarification.

Reviewer: To start with, the Abstract. Include the number of subjects, location (Greece), and calendar time (year).

Response: we would like to thanks the reviewer for the comment. We add the phrase in abstract section as follow: “The present study was conducted on dental patients visiting four private dental offices in the city of Larissa in Central Greece during the period October-December 2021”. Lines 28-30

Reviewer: The sample size, viz., 200, is modest. It is large enough to make some inferences, especially with a new, rapidly developing topic such as COVID. Remove all decimal places, especially in %s. The sample size does not warrant that level of precision.

Response: we would like to thanks the reviewer for the constructive comment. We removed all decimal places. About the sample size refer by the authors as a significant limitation of the study in the limitations “lines 237-245”.

Reviewer: Methods. The investigators did an excellent job in development of the “instrument/questionnaire”. They should state that it was a brief, demographics and 8 questions on attitude/perceptions. Many will presume a much longer survey.

Below are the 8 questions from Table 2 – I presume comprise the non-demographic portion of the survey. There is a disconnect in the form of the question for #3,4 6, where Table 2 has likert response options. The way the questions are worded in the table (and below) only a yes/no would be appropriate response options. I suspect that it is a translation issue. I presume that the questionnaire was completed in Greek and translated for this publication. Consider the rewording of #3,4,6 below – IF that is how the questions read originally.

Do you agree/disagree with each of the following statements:

3) The dentist's role is important in dealing with the Covid-19 pandemic.

4) The information you received from the dentist about the Covid-19 pandemic sufficient.

5) You consider for your appointments whether the dentist follows all health protocols against

Covid-19 according to guidelines of public health authorities.

Response: thanks, the reviewer for the constructive comment. Yes, the questionnaire was completed in Greek and translated for this publication. We add the reviewer suggestions about the questionnaire “brief, demographics and 8 questions on attitude/perceptions” lines 88-96.

We agree with the suggestions of the reviewer and it is a translation issue. We modified the questions according to reviewer suggestions.

Reviewer: Instrument – Based on Table 2 1. Did you undergo teeth cleaning during the last year? Y/N 2. Did you cancel planned prosthetic work (dentures, implants, crown placements) due to the fear of the transmission of Covid-19? Y/N 3. Do you think that the dentist's role is important in dealing with the Covid-19 pandemic? LIKERT 4. Was the information you received from the dentist about the Covid-19 pandemic sufficient? LIKERT 5. Has Covid-19 discouraged you to keeping your scheduled dental appointments? Y/N 6. Do you consider for your appointments that the dentist follows all health protocols against Covid-19 according to guidelines of public health authorities’? LIKERT 7. If you knew that the dentist he had fallen ill from Covid-19 and fully recovered, would it affect your visit to the dental office? Y/N 8. Would you feel safe knowing that your dentist was vaccinated against the Covid-19 disease? Y/N

Response: thanks, the reviewer for the comment. The questions 1,2,5,7,8 is Y/N and the questions 3,4,6 is LIKERT.

Reviewer: Statistics. Univariate stats are means, sd, etc. For age, I suggest giving mean (sd) in text not table. Also give median and inter-quartile range (IQR). The latter will give the reader a much better understanding of the distribution of ages among the participants. Chi-square and t-tests and bivariate analyses, not univariate descriptors

Response: thanks, the reviewer for the comment. We deleted age and SD from the table accordingly. About the distribution of ages, we preferred to present the sample of the study in the present form because of the dissimilarity of the age decades. The vast majority of the participants were up the 40 years and we had small contribution in ages 18-35.

Reviewer: Results. The results are hard to follow, fully grasp. One of the objectives, in fact the primary objective, is to ascertain the effect of a health care worker (dentist) being vaccinated against COVID on patients seeking care from them. In the logistic regression model the outcome is “decision of patients to attend to the appointment with the dentist”. What number is this? Is it the 12% (N=36) would feel safe knowing their dentist was vaccinated? Or was it the 22% (N=67) who would be affected if their dentist had had COVID but had recovered. Whatever it is, it needs to be explicitly stated and put in the title of the table. Also, before doing any model, the individual, or bivariate analyses should be examined and presented. I suggest adding columns to tables 1 and 2 that indicate the outcome. Education should be in table 1, not as a chart. Age as a categorical variable to better present any association with the outcome. Below is a common table format.

Response: thanks, the reviewer for the comment. We deleted figure one and the education level transferred to table 1.

 About the question in regression analysis “decision of patients to attend to the appointment with the dentist”. The outcome variable of the model is “has Covid-19 discouraged you to keeping your scheduled dental appointments?” no (67%). We added the suggested info in a new table for baseline model and reduced. Regarding bivariate analysis statistically significant results are presented in the corresponding section.

Reviewer 2 Report

In the communication entitled “Dental patients' perceptions of their perceived susceptibility to COVID-19 infection depend on dentists' previous infection and vaccination status, which further affect their compliance with healthcare services.” the authors investigated the correlation between the perceived susceptibility of patients with dental disorders regarding the implementation of protective measures. The article in is line with journal’s aim, and the Authors have well revised several issues; however, I ask authors to add some key concepts.

-       The title is redundant, please specify the type of study and shorten it.

-       Is this a communication? I think there was an error in choosing the type of article.

-       The abstract section is too confusing, please rewrite the background part (moreover the aim of the study is not easily identifiable)

-        The introduction section must be implemented, starting from how Aerosol-generating procedures (AGPs) in dental practice remain a health concern, since aerosols produced during clinical procedures can be contaminated with microorganisms, which can cause respiratory health effects or transmit diseases bidirectionally among dental professionals and patients, such as Sars-Cov-2 (please see and discuss https://doi.org/10.3390/ijerph18147472

  • What are the limits of the study?
  • Conclusions cannot be reduced to a sentence: you must improve them highlighting the limits and the future insights pointed out from this article.
  • The formatting of the references is not correct, please check the journal instructions for authors
  • Several moderate typos are present in the text, please, amend, according to

According to this Reviewer’s consideration, novelty and quality of the paper, publication of the present manuscript is recommended after minor revision.

Author Response

Reviewer 2.

In the communication entitled “Dental patients' perceptions of their perceived susceptibility to COVID-19 infection depend on dentists' previous infection and vaccination status, which further affect their compliance with healthcare services.” the authors investigated the correlation between the perceived susceptibility of patients with dental disorders regarding the implementation of protective measures. The article in is line with journal’s aim, and the Authors have well revised several issues; however, I ask authors to add some key concepts.

 Reviewer: The title is redundant, please specify the type of study and shorten it.

Answer: we would like to thank the reviewer. We chanced the title to “Factors for compliance of dental patients in primary health care services during the pandemic” and we add the type of study as cross-sectional study.

Reviewer: Is this a communication? I think there was an error in choosing the type of article.

Answer: we would like to thank the reviewer for the comment. We choose the type of article "communication" under the status of the number of participants of the study and the possibility who not depict the opinions of general population.

Reviewer:  The abstract section is too confusing, please rewrite the background part (moreover the aim of the study is not easily identifiable).

Answer: we would like to thank the reviewer for the comment. We modified the background part accordingly “The compliance of dentists of good hygiene practices during the COVID-19 pandemic is essential to minimize the transmission of SARS-CoV-2 strains and the pandemic, has significantly interrupted the provision of oral health care to many individuals” lines 21-23  and the aim of the study as follow “  We aimed to examine in a cross- sectional study factors affecting dental patients’ and compliance in primary dental health settings during the pandemic period” lines 27-28 .  

Reviewer:    The introduction section must be implemented, starting from how Aerosol-generating procedures (AGPs) in dental practice remain a health concern, since aerosols produced during clinical procedures can be contaminated with microorganisms, which can cause respiratory health effects or transmit diseases bidirectionally among dental professionals and patients, such as Sars-Cov-2 (please see and discuss https://doi.org/10.3390/ijerph18147472

Answer: we would like to thank the reviewer for the constructive comment. We add the phrase “The way of transmission by aerosol-generating procedures (AGPs) is different from droplet transmission as it refers to the presence of microbes within droplet nuclei, which are generally considered to be particles <5μm in diameter, can remain in the air for long periods of time and be transmitted to others over distances greater than 1 m. Aerosol generating procedures (AGPs) in dental practice remain a health concern, since aerosols produced during clinical procedures can be contaminated with microorganisms, which can cause respiratory health effects or transmit diseases bidirectionally among dental professionals and patients” and we add the new reference number (4) https://doi.org/10.3390/ijerph18147472.  In lines 58-65.

Reviewer:     What are the limits of the study?

Answer: we would like to thanks the reviewer for the commend. At the end of discussion section included a plethora of limitations of the present study.  Especially in lines 234-242“The present study has several limitations. It was carried out during the third wave of the pandemic when there had already been some degree of familiarization of patients and dentists with the increased protection measures. The present study was descriptive, and due to its nature is not possible for the authors to provide the reader with causal associations between investigated risk factors and compliance with primary healthcare services during the pandemic. Due to the questionnaire-based structure of our study, recall or information bias may have occurred. The convenience sample of the participants was another limitation, and the small number of dental office participants needs to be representative of the dentist community” we believe that indicate the limitations of the present study to be a full article.

Reviewer:        Conclusions cannot be reduced to a sentence: you must improve them highlighting the limits and the future insights pointed out from this article.

Answer: thanks, the reviewer for the comment. About the limits please see the preview answer. We reform the conclusions section accordingly.

Reviewer:    The formatting of the references is not correct, please check the journal instructions for authors

Answer: thanks, the reviewer for the comment we modified the references according to journal instructions.

Reviewer:            Several moderate typos are present in the text, please, amend, according to

Answer: Thanks, the reviewer for the comment. We would like to inform that the author (4) she is a native English speaker. 

According to this Reviewer’s consideration, novelty and quality of the paper, publication of the present manuscript is recommended after minor revision

Round 2

Reviewer 1 Report

The authors have responded well to my original comments with these exceptions. 1) Put the sample size in the abstract, e.g., ... conducted on 300 dental patients ..... Insert '300'.  2) Give more information on age range of patients, with mean and sd, give median and IQR, or perhaps chart like education was originally presented. 3) This is the most important - the outcome is still NOT presented in the table. 4) No more than 2 decimal places for odds ratios and 95% CIs in table 3.

Author Response

Response to Reviewer Comments

Point 1: The authors have responded well to my original comments with these exceptions. 1) Put the sample size in the abstract, e.g., ... conducted on 300 dental patients ..... Insert '300'. 

Response 1: We would like to thanks the reviewer for the suggestion. We modified the text accordingly.

Point 2: Give more information on age range of patients, with mean and sd, give median and IQR, or perhaps chart like education was originally presented.

Response 2: We would like to thanks the reviewer for the comment. We added in text (line 111) the range of participants age. Furthermore, we added the mean age and IQR. About the presentation of age on table we followed the initial suggestions of 1st reviewer and erased the ages from the table of demographics.

Point 3: This is the most important - the outcome is still NOT presented in the table.

Response 3: We would like to thanks the reviewer for the comment. The table 2 presentts outcomes souch (the information regarding COVID-19 that dentist was given to their clients, the feeling of safety of the patients if they knew their dentist was vaccinated against COVID-19  and we report that dentists had adequate knowledge, attitude, and practice on infection control protocols against COVID-19). In addition the outcome is now stated in the heading of the tample 3.

Point 4: No more than 2 decimal places for odds ratios and 95% CIs in table 3.

Response 4: We modified the table accordingly.
